# MODELING SEQUENTIAL SENTENCE RELATION TO IMPROVE CROSS-LINGUAL DENSE RETRIEVAL

**Shunyu Zhang**[1,*]**, Yaobo Liang**[1]**, Ming Gong**[2]**, Daxin Jiang**[2]**, Nan Duan**[1]
[1]Microsoft Research Asia, [2]Microsoft STC Asia
`shunyuzh@foxmail.com, {yalia, migon, djiang, nanduan}@microsoft.com`

## ABSTRACT

Recently multi-lingual pre-trained language models (PLM) such as mBERT and XLM-R have achieved impressive strides in cross-lingual dense retrieval. Despite its successes, they are general-purpose PLM while the multilingual PLM tailored for cross-lingual retrieval is still unexplored. Motivated by an observation that the sentences in parallel documents are approximately in the same order, which is universal across languages, we propose to model this sequential sentence relation to facilitate cross-lingual representation learning. Specifically, we propose a multi-lingual PLM called masked sentence model (MSM), which consists of a sentence encoder to generate the sentence representations, and a document encoder applied to a sequence of sentence vectors from a document. The document encoder is shared for all languages to model the universal sequential sentence relation across languages. To train the model, we propose a masked sentence prediction task, which masks and predicts the sentence vector via a hierarchical contrastive loss with sampled negatives. Comprehensive experiments on four cross-lingual retrieval tasks show MSM significantly outperforms existing advanced pre-training models, demonstrating the effectiveness and stronger cross-lingual retrieval capabilities of our approach.

## 1 INTRODUCTION

Cross-lingual retrieval (also including multi-lingual retrieval) is becoming increasingly important as new texts in different languages are being generated every day, and people query and search for the relevant documents in different languages (Zhang et al., 2021b; Asai et al., 2021a). This is a fundamental and challenging task and plays an essential part in real-world search engines, for example, Google and Bing search which serve hundreds of countries across diverse languages. In addition, it's also a vital component to solve many cross-lingual downstream problems, such as open-domain question answering (Asai et al., 2021a) or fact checking (Huang et al., 2022).

With the rapid development of deep neural models, cross-lingual retrieval has progressed from translation-based methods (Nie, 2010), cross-lingual word embeddings (Sun & Duh, 2020), and now to dense retrieval built on the top of multi-lingual pre-trained models (Devlin et al., 2019; Conneau et al., 2019). Dense retrieval models usually adopt pretrained models to encode queries and passages into low-dimensional vectors, so its performance relies on the representation quality of pretrained models, and for multilingual retrieval it also calls for cross-lingual capabilities.

Models like mBERT (Devlin et al., 2019), XLMR (Conneau et al., 2019) pre-trained with masked language model task on large multilingual corpora, have been applied widely in cross-lingual retrieval (Asai et al., 2021a;b; Shi et al., 2021) and achieved promising performance improvements. However, they are general pre-trained models and not tailored for dense retrieval. Except for the direct application, there are some pre-trained methods tailored for monolingual retrieval. Lee et al. (2019) and Gao & Callan (2021) propose to perform contrastive learning with synthetic query-document pairs to pre-train the retriever. They generate pseudo pairs either by selecting a sentence and its context or by cropping two sentences in a document. Although showing improvements, these approaches have only been applied in monolingual retrieval and the generated pairs by hand-crafted

---

*Work done during internship at Microsoft Research Asia.

rules may be low-quality and noisy. In addition, learning universal sentence representations across languages is more challenging and crucial than monolingual, so better multilingual pre-training for retrieval needs to be explored.

In this paper, we propose a multilingual PLM to leverage sequential sentence relation across languages to improve cross-lingual retrieval. We start from an observation that the parallel documents should each contain approximately the same sentence-level information. Specifically, the sentences in parallel documents are approximately in the same order, while the words in parallel sentences are usually not. It means the sequential relation at sentence-level are similar and universal across languages. This idea has been adopted for document alignment (Thompson & Koehn, 2020; Resnik, 1998) which incorporates the order information of sentences. Motivated by it, we propose a novel Masked Sentence Encoder (MSM) to learn this universal relation and facilitate the isomorphic sentence embeddings for cross-lingual retrieval. It consists of a sentence encoder to generate sentence representations, and a document encoder applied to a sequence of sentences in a document. The document encoder is shared for all languages and can learn the sequential sentence relation that is universal across languages. In order to train MSM, we adopt a sentence-level masked prediction task, which masks the selected sentence vector and predicts it using the output of the document encoder. Distinct from MLM predicting tokens from pre-built vocabulary, we propose a hierarchical contrastive loss with sampled negatives for sentence-level prediction.

We conduct comprehensive experiments on 4 cross-lingual dense retrieval tasks including Mr. TyDi, XOR Retrieve, Mewsli-X and LAReQA. Experimental results show that our approach achieves state-of-the-art retrieval performance compared to other advanced models, which validates the effectiveness of our MSM model in cross-lingual retrieval. Our in-depth analysis demonstrates that the cross-lingual transfer ability emerges for MSM can learn the universal sentence relation across languages, which is beneficial for cross-lingual retrieval. Furthermore, we perform ablations to motivate our design choices and show MSM works better than other counterparts.

## 2 RELATED WORK

**Multi-lingual Pre-trained Models.** Recently the multilingual pre-trained models (Lample & Conneau, 2019; Conneau et al., 2019; Huang et al., 2019) have empowered great success in different multilingual tasks (Liang et al., 2020; Hu et al., 2020). Multilingual BERT (Devlin et al., 2019) is a transformer model pre-trained on Wikipedia using the multi-lingual masked language model (MMLM) task. XLM-R (Conneau et al., 2019) further extends the corpus to a magnitude more web data with MMLM. XLM (Lample & Conneau, 2019) proposes the translation language model (TLM) task to achieve cross-lingual token alignment. Unicoder (Huang et al., 2019) presents several pre-training tasks upon parallel corpora and ERNIE-M (Ouyang et al., 2021) learns semantic alignment by leveraging back translation. XLM-K (Jiang et al., 2022) leverages the multi-lingual knowledge base to improve cross-lingual performance on knowledge-related tasks. InfoXLM (Chi et al., 2021) and HiCTL (Wei et al., 2020) encourage bilingual alignment via InfoNCE based contrastive loss. These models usually focus on cross-lingual alignment leveraging bilingual data, while it's not fit for cross-lingual retrieval that calls for semantic relevance between query and passage. There is few explore on how to improve pre-training tailored for cross-lingual retrieval, which is exactly what our model addresses.

**Cross-lingual Retrieval.** Cross-lingual (including multi-lingual) retrieval is becoming increasingly important in the community and impacting our lives in real-world applications. In the past, multi-lingual retrieval relied on community-wide datasets at TREC, CLEF, and NCTIR, such as CLEF 2000-2003 collection (Ferro & Silvello, 2015). They usually comprise a small number of queries (at most a few dozen) with relevance judgments and only for evaluation, which are insufficient for dense retrieval. Recently, more large scale cross-lingual retrieval datasets (Zhang et al., 2021b; Ruder et al., 2021) have been proposed to promote cross-lingual retrieval research, such as Mr. TyDi (Asai et al., 2021a) proposed in open-QA domain, Mewsli-X (Ruder et al., 2021) for news entity retrieval, etc.

The technique of the cross-lingual retrieval field has progressed from translation-based methods (Nie, 2010; Shi et al., 2021) to cross-lingual word embeddings by neural models (Sun & Duh, 2020), and now to dense retrieval built on the top of multi-lingual pre-trained models (Devlin et al., 2019; Conneau et al., 2019). Asai et al. (2021a;b) modify the bi-encoder retriever to be equipped with mBERT, which plays an essential part in the open-QA system, and Zhang et al. (2022b) explore

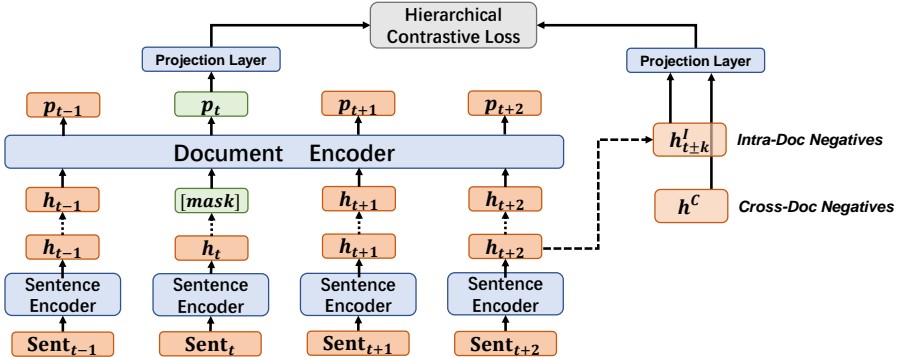

Figure 1: The general framework of masked sentence model (MSM), which has a hierarchical model architecture including the sentence encoder and the document encoder. The masked sentence prediction task predicts the masked sentence vector $p_t$, given the original vector $h_t$ as the positive anchor, via a hierarchical contrastive loss.

the impact of data and model. However, most of the existing work focuses on fine-tuning a specific task, while ours targets pre-training and conducts evaluations on diverse benchmarks. There also exist some similarity-specialized multi-lingual models (Litschko et al., 2021), trained with parallel or labeled data supervision. LASER (Artetxe & Schwenk, 2019) train a seq2seq model on large-scale parallel data and LaBSE (Feng et al., 2022) encourage bilingual alignment via contrastive loss. m-USE (Yang et al., 2019) is trained with mined QA pairs, translation pairs and SNLI corpus. Some others also utilize distillation (Reimers & Gurevych, 2020; Li et al., 2021), adapter (Pfeiffer et al., 2020; Litschko et al., 2022), siamese learning (Zhang et al., 2021c). Compared to them, MSM is unsupervised without any parallel data, which is more simple and effective (Artetxe et al., 2020b), and can also be continually trained with these bi-lingual tasks.

**Dense Retrieval.** Dense retrieval (Karpukhin et al., 2020; Lee et al., 2019; Xiong et al., 2020; Zhang et al., 2022a) (usually monolingual here) typically utilizes bi-encoder model to encode queries and passages into low-dimensional representations. Recently there have been several directions explored in the pre-training tailored for dense retrieval: utilizing the hyperlinks between the Wikipedia pages (Ma et al., 2021; Zhou et al., 2022), synthesizing query-passage datasets for pre-training (Oğuz et al., 2021; Reddy et al., 2021), and auto-encoder-based models that force the better representations (Lu et al., 2021; Ma et al., 2022). Among them, there is a popular direction that leverage the correlation of intra-document text pairs for the pre-training. Lee et al. (2019) and Chang et al. (2020) propose Inverse Close Task (ICT) to treat a sentence as pseudo-query and the concatenated context as the pseudo-passage for contrastive pre-training. Another way is cropping two sentence spans (we call them CROP in short) from a document as positive pairs (Giorgi et al., 2021; Izacard et al., 2021a), including Wu et al. (2022a); Iter et al. (2020) that use two sentences and Gao & Callan (2021) that adopts two non-overlapping spans. The most relevant to ours are ICT and CROP, which generate two views of a document for contrastive learning. However, the correlation of the pseudo pair is coarse-granular and even not guaranteed. In contrast, ours utilizes a sequence of sentences in a document and models the universal sentence relation across languages via an explicit document encoder, resulting in better representation for cross-lingual retrieval.

# 3 METHODOLOGY

## 3.1 HIERARCHICAL MODEL ARCHITECTURE

In this section, we first present the hierarchical model architecture. As illustrated in Figure.1, our Masked Sentence Encoder (MSM) has a hierarchical architecture that contains the Sentence Encode and the Document Encoder. The document encoder is applied to the sentence vectors generated by the sentence encoder from a sequence of sentences in a document.

**Sentence Encoder.** Given a document containing a sequence of sentences $\mathcal{D} = (S_1, S_2, ..., S_N)$ in which $S_i$ denote a sentence in document, and each sentence contains a list of words. As shown

in 1, the sentence encoder extracts the sentence representations for the sequence in a document, and the document encoder is to model the sentence relation and predict the masked sentences. First, we adopt a transformer-based encoder as our sentence encoder. Then as usual the sentence is passed through the embedding layer and transformer layers, and we take the last hidden state of the CLS token as the sentence representation. Note that all the sentence encoders share the parameters and we can get $N$ sentence vectors $\mathcal{D}_H = (h_1, h_2, ..., h_N)$ respectively. In this task, we just encoder the complete sentences without the mask to get thorough sentence representations.

**Document Encoder.** Then the sentence-level vectors run through the document encoder, which has similar transformer-based architecture. Considering the sentences have sequential order in a document, the sentence position is also taken into account and it doesn't have token type embedding. After equipped with sentence position embeddings, we encode them through document encoder layers to get document-level context aware embeddings $\mathcal{D}_P = (p_1, p_2, ..., p_N)$. In order to train our model, we apply the sentence-level mask to the sentence vectors for our masked sentence prediction task. Specifically, $\mathcal{D}_H = (h_1, h_2, ..., h_N)$ are the original sentence vectors, and we mask selected sentence vector $h_t$ to $[mask]$ token and keep other the original ones. The original sentence vector $h_t$ is seen as the positive anchor for the output vector $p_t$ of document encoder corresponding to $[mask]$. Considering a document that contains $N$ sentences, we mask each sentence in turn and keep the others the original, to get $N$ pairs of $p_t$ and $h_t$. It is effective to get as many samples as possible at the same time for efficient training. Since the length of document encoder's input is not long (for the number of sentences in a document is not long) and our document encoder is also shallow, it makes our approach efficient without much computation.

There are some models also adopting a hierarchical transformer-based model (Santra et al., 2021). For example, HiBERT (Zhang et al., 2019) uses a multi-level transformers for document summarization, while it applies the mask to the words with a decoder for autoregressive pre-training. Poolingformer (Zhang et al., 2021a) proposes a two-level pooling attention schema for long document but can't be applied for retrieval. They mainly adopt token-level MLM and targets document understanding, while ours focuses on masked sentence prediction and is directed at cross-lingual retrieval.

### 3.2 MASKED SENTENCE PREDICTION TASK

In order to model the sentence relation, we propose a masked sentence prediction task that aligns masked sentence vectors $p_t$ with corresponding original $h_t$ via the hierarchical contrastive loss. Distinct from Masked Language Model which can directly compute cross-entropy loss between masked tokens and pre-built vocabulary, our model lacks a sentence-level vocabulary. Here we propose a novel hierarchical contrastive loss on sentence vectors to address it. Contrastive learning has been shown effective in sentence representation learning (Karpukhin et al., 2020; Gao et al., 2021), and our model modifies the typical InfoNCE loss (Oord et al., 2018) to a hierarchical contrastive loss for the masked sentence prediction task. As shown in Figure.1, for masked sentence vectors $p_t$, the positive anchor is original $h_t$ and we collect two categories of negatives: (a) Cross-Doc Negatives are the sentence vectors from different documents, i.e. $h_k^{\mathcal{C}}$, which can be seen as random negatives as usual. (b) Intra-Doc Negatives are the sentence vectors in a same document generated by sentence encoder, i.e. $h_j^{\mathcal{I}}, j \neq t$. Then the masked sentence vectors $p_t$ with them are passed through the projection layer, and the output vectors are involved in the hierarchical contrastive loss as:

$$
\begin{aligned}
&\mathcal{L}_{msm}(p_t, \{h_t^{\mathcal{I}}, h_1^{\mathcal{I}}, \ldots, h_{|\mathcal{I}|}^{\mathcal{I}}, h_1^{\mathcal{C}}, \ldots, h_{|\mathcal{C}|}^{\mathcal{C}}\}) \\
&= -\log \frac{e^{\mathrm{sim}\left(p_t, h_t^{\mathcal{I}}\right)}}{e^{\mathrm{sim}\left(p_t, h_t^{\mathcal{I}}\right)-} + \sum_{j=1, j \neq t}^{|\mathcal{I}|} e^{\mathrm{sim}\left(p_t, h_j^{\mathcal{I}}\right)-\mu\alpha} + \sum_{k=1}^{|\mathcal{C}|} e^{\mathrm{sim}\left(p_t, h_k^{\mathcal{C}}\right)}}
\end{aligned}
\tag{1}
$$

In the previous study (Gao & Callan, 2021), two sampled views or sentences of the same document are often seen as a positive pair to leverage their correlation. However, it limits the representation capability for it encourages the alignment between two views, just as a coarse-grained topic model (Yan et al., 2013). In contrast, we treat them as Intra-Doc Negatives, which could help the model to distinguish sentences from the same document to learn fine-grained representations. The intra-doc samples usually have closer semantic relation than cross-doc ones and directly treating them as negatives could hurt the uniformity of embedding space. To prevent this negative impact,

we set the dynamic bias subtracted from their similarity scores. As seen in Eq.1, the dynamic bias is $-\mu\alpha$ in which $\mu$ is a hyper-parameter and $\alpha$ is computed as:

$$\alpha = \left( \frac{\sum_{j=1, j \neq t}^{|\mathcal{I}|} \text{sim}\left(p_t, h_j^{\mathcal{I}}\right)}{|\mathcal{I}| - 1} - \frac{\sum_{k=1}^{|\mathcal{C}|} \text{sim}\left(p_t, h_k^{\mathcal{C}}\right)}{|\mathcal{C}|} \right) .detach() \tag{2}$$

It represents the gap between the average similarity score of Intra-Doc Negatives and them from Cross-Doc Negatives. Subtracting the dynamic bias can tune the high similarity of intra-doc negatives to the level of cross-doc negatives, which can also be seen as interpolation to generate soft samples. Note that we only use the value but do not pass the gradient, so we adopt the detach function after computation. Our experimental result in Sec.5.4 validates that the hierarchical contrastive loss is beneficial for representation learning in our model.

Considering the expensive cost of pre-training from scratch, we initialize the sentence encoder with pre-trained XLM-R weight and solely the document encoder from scratch. To prevent gradient back propagated from the randomly initialized document encoder from damaging sentence encoder weight, we adopt MLM task to impose a semantic constraint. Therefore our total loss consists of a token-level MLM loss and a sentence-level contrastive loss:

$$\mathcal{L} = \mathcal{L}_{msm} + \mathcal{L}_{mlm} \tag{3}$$

After pre-training, we discard the document encoder and leave the sentence encoder for fine-tuning. In fact, the document encoder in our MSM plays as a bottleneck (Li et al., 2020): the sentence encoder press the sentence semantics into sentence vectors, and the document encoder leverage the limited information to predict the masked sentence vector, thus enforcing an information bottleneck on the sentence encoder for better representations. It also coincides with the recent works utilizing similar bottleneck theory for better text encoders (Lu et al., 2021; Liu & Shao, 2022). By the way, the sentence encoder has the same architecture as XLMR, which ensures a fair comparison.

## 4 EXPERIMENTS SETUP

### 4.1 EVALUATION DATASETS

We evaluate our model with other counterparts on 4 popular datasets: Mr. TyDi is for query-passage retrieval, XOR Retrieve is cross-lingual retrieval for open-domain QA, Mewsli-X and LAReQA are for language-agnostic retrieval. Mr. TyDi (Zhang et al., 2021b) aims to evaluate cross-lingual passage retrieval with dense representations. Given a question in language $L$, the model should retrieve relevant texts in language $L$ that can respond to the query. XOR Retrieve (Asai et al., 2021a) is proposed for multilingual open-domain QA, and we take its sub-task XOR-Retrieve: given a question in $L$ (e.g., Korean), the task is to retrieve English passages that can answer the query. Mewsli-X is built on top of Mewsli (Botha et al., 2020) and we follow the setting of XTREME-R (Ruder et al., 2021), in which it consisting of 15K mentions in 11 languages. LAReQA (Roy et al., 2020) is a retrieval task that each query has target answers in multiple languages, and models require retrieving all correct answers regardless of language. More details refer to Appendix A.1.

### 4.2 IMPLEMENTATION DETAILS

For the pre-training stage, we adopt transformer-based sentence encoder initialized from the XLMR weight, and a 2-layers transformer document encoder trained from scratch. We use a learning rate of 4e-5 and Adam optimizer with a linear warm-up. Following Wenzek et al. (2019), we collect a clean version of Common Crawl (CC) including a 2,500GB multi-lingual corpus covering 108 languages, which adopt the same pre-processing method as XLMR (Conneau et al., 2019). Note that we only train on CC without any bilingual parallel data, in an unsupervised manner. To limit the memory consumption during training, we limit the length of each sentence to 64 words (longer parts are truncated) and split documents with more than 32 sentences into smaller with each containing at most 32 sentences. The rest settings mainly follow the original XLMR's in FairSeq. We conduct pre-training on 8 A100 GPUs for about 200k steps.

Table 1: Retrieval performance comparison on four benchmark datasets. The best performing models are marked bold and the results unavailable are left blank. * means the results are borrowed from published papers: † from Zhang et al. (2022b), ‡ from Asai et al. (2021a), § from Ruder et al. (2021), while others are evaluated using the same pipeline by us for a fair comparison.

| Model | Mr. TyDi | | XOR Retrieve | | Mewsli-X | LAReQA |
|---|---|---|---|---|---|---|
| | MRR@100 | R@100 | R@2k | R@5k | mAP@20 | mAP@20 |
| *Cross-lingual zero-shot transfer (models are fine-tuned on English data)* | | | | | | |
| mBERT* | 34.4† | 73.4† | - | - | 38.6‡ | 21.6‡ |
| mBERT | 33.0 | 69.7 | 31.6 | 42.3 | 39.0 | 24.5 |
| XLMR | 37.7 | 72.7 | 30.6 | 39.3 | 39.7 | 29.3 |
| MSM | **44.7** | **78.6** | **34.9** | **44.7** | **41.5** | **33.5** |
| *Multi-lingual fine-tune (models are fine-tuned on the multi-lingual data if available)* | | | | | | |
| mBERT* | 59.1† | 87.1† | 38.8§ | 48.0§ | - | - |
| mBERT | 57.6 | 87.9 | 48.3 | 57.0 | - | - |
| XLMR | 58.5 | 87.8 | 45.1 | 53.9 | - | - |
| MSM | **60.5** | **89.0** | **48.6** | **57.3** | - | - |

For the fine-tuning stage, we mainly follow the hyper-parameters of the original paper for the Mr. TyDi and XOR Retrieve tasks separately. And for Mewsli-X and LAReQA tasks, we mainly follow the settings of XTREME-R using its open-source codebase. Note that we didn't tune the hyper-parameters and mainly adopted the original settings using the same pipeline for a fair comparison. More details of fine-tuning hyper-parameters refer to Appendix.A.1.

## 5 EXPERIMENT RESULTS

### 5.1 EVALUATION SETTINGS

**Cross-lingual Zero-shot Transfer.** This setting is most widely adopted for the evaluation of multilingual scenarios with English as the source language, as many tasks only have labeled train data available in English. Concretely, the models are fine-tuned on English labeled data and then evaluated on the test data in the target languages. It also facilitates evaluation as models only need to be trained once and can be evaluated on all other languages. For Mr. TyDi dataset, the original paper adopt the Natural Questions data (Kwiatkowski et al., 2019) for fine-tuning while later Zhang et al. (2022b) suggests fine-tuning on MS MARCO for better results, so we fine-tune on MARCO when compared with best-reported results and on NQ otherwise. For XOR Retrieve, we fine-tune on NQ dataset as the original paper (Asai et al., 2021a). For Mewsli-X and LAReQA, we follow the settings in XTREME-R, where Mewsli-X on a predefined set of English-only mention-entity pairs and LAReQA on the English QA pairs from SQuAD v1.1 train set.

**Multi-lingual Fine-tune.** For the tasks where multi-lingual training data is available, we additionally compare the performance when jointly trained on the combined training data of all languages. Following the setting of Mr. TyDi and XOR Retrieve, we pre–fine-tune models as in Cross-lingual Zero-shot Transfer and then fine-tune on multi-lingual data if available. For the Mewsli-X and LAReQA, there is no available multi-lingual labeled data.

### 5.2 MAIN RESULTS

In this section, we evaluate our model on diverse cross-lingual and multi-lingual retrieval tasks and compare it with other strong pre-training baselines. Multilingual BERT (Devlin et al., 2019) pre-trained on multilingual Wikipedia using MLM and XLMR (Conneau et al., 2019) extend to a magnitude more web data for 100 languages.

We report the main results in Table.1, and provide more detailed results in Appendix A.2. The effectiveness of our multilingual pre-training method appears clearly as MSM achieves significantly better performance than its counterparts. (1) No matter whether in the cross-lingual zero-shot transfer or multi-lingual fine-tune settings, the performance trend is consistent and MSM achieves impressive

Table 2: Performance comparison on Mr. TyDi across languages in the cross-lingual zero-shot transfer setting, where all the models are fine-tuned on MS MARCO data. - means it doesn't support BN and TE languages and the average is for the supported languages.

| Method | Metrics | AR | BN | EN | FI | ID | JA | KO | RU | SW | TE | TH | AVG |
|---|---|---|---|---|---|---|---|---|---|---|---|---|---|
| *Similarity-specialized multi-lingual encoders (with parallel data or labeled data supervision)* | | | | | | | | | | | | | |
| DistilmBERT | MRR@100 | 40.8 | - | 29.9 | 26.7 | 39.7 | 27.0 | 32.2 | 29.4 | 22.0 | - | 26.5 | 30.5 |
| | Recall@100 | 79.7 | - | 71.0 | 64.1 | 79.7 | 65.0 | 64.4 | 62.6 | 48.2 | - | 60.9 | 66.2 |
| InfoXLM | MRR@100 | 48.2 | 50.6 | 30.1 | 29.0 | 39.9 | 30.1 | 34.8 | 35.0 | 38.9 | 51.7 | 50.9 | 39.9 |
| | Recall@100 | 81.2 | 83.8 | 72.2 | 65.8 | 75.9 | 68.1 | 70.0 | 72.7 | 69.3 | 81.0 | 87.1 | 75.2 |
| LaBSE | MRR@100 | 50.1 | 52.3 | 29.7 | 41.3 | 48.3 | 27.6 | 33.4 | 37.3 | 54.6 | 56.7 | 43.6 | 43.2 |
| | Recall@100 | 83.0 | 85.6 | 71.4 | 80.2 | 86.1 | 63.2 | 67.0 | 74.3 | 86.7 | 89.4 | 81.9 | 79.0 |
| *Unsupervised pre-train with multi-lingual corpus (Wiki or CC)* | | | | | | | | | | | | | |
| mBERT | MRR@100 | 45.3 | 38.8 | 29.1 | 28.7 | 35.5 | 29.6 | 29.8 | 32.6 | 27.8 | 40.8 | 25.3 | 33.0 |
| | Recall@100 | 77.8 | 84.7 | 73.8 | 65.8 | 72.9 | 68.4 | 59.9 | 71.1 | 56.4 | 76.8 | 59.2 | 69.7 |
| XLMR | MRR@100 | 43.8 | 41.2 | 29.3 | 33.2 | 45.4 | 27.0 | 33.4 | 32.2 | 35.3 | 44.5 | 49.7 | 37.7 |
| | Recall@100 | 77.4 | 81.5 | 68.5 | 72.2 | 81.8 | 61.2 | 66.5 | 66.4 | 63.9 | 75.5 | 85.4 | 72.8 |
| XLMR-Long | MRR@100 | 43.9 | 44.7 | 27.2 | 31.6 | 44.2 | 28.5 | 34.1 | 30.9 | 31.0 | 49.5 | 48.2 | 37.6 |
| | Recall@100 | 75.8 | 82.0 | 68.5 | 71.1 | 81.6 | 62.9 | 64.9 | 64.7 | 58.9 | 80.0 | 85.9 | 72.4 |
| CROP | MRR@100 | 46.2 | 39.9 | 27.7 | 34.0 | 47.0 | 26.2 | 32.1 | 31.8 | 41.5 | 55.9 | 46.3 | 38.9 |
| | Recall@100 | 80.3 | 84.2 | 70.9 | 73.8 | 85.5 | 63.1 | 68.1 | 70.4 | 72.0 | 86.3 | 85.9 | 76.4 |
| MSM | MRR@100 | 51.6 | 53.0 | 31.6 | 39.4 | 50.5 | 32.0 | 36.8 | 37.2 | 43.4 | 62.6 | 53.5 | 44.7 |
| | Recall@100 | 83.0 | 83.8 | 73.9 | 77.9 | 85.7 | 67.5 | 70.3 | 71.4 | 73.0 | 89.8 | 88.2 | 78.6 |

Table 3: Cross-lingual zero-shot transfer results on XOR Retrieve task. We report R@2k and R@5 metrics on the test sets. All compared methods are unsupervised pre-trained models.

| Method | Metrics | AR | BN | FI | JA | KO | RU | TE | AVG |
|---|---|---|---|---|---|---|---|---|---|
| mBERT | R@2k | 31.1 | 26.6 | 38.5 | 32.4 | 38.6 | 24.9 | 29.1 | 31.6 |
| | R@5k | 44.1 | 36.2 | 48.1 | 41.5 | 48.1 | 38.4 | 39.5 | 42.3 |
| XLMR | R@2k | 39.9 | 25.7 | 41.1 | 27.8 | 31.9 | 22.4 | 25.2 | 30.6 |
| | R@5k | 49.6 | 34.9 | 50.0 | 34.9 | 41.8 | 30.4 | 34.0 | 39.3 |
| CROP | R@2k | 45.4 | 32.9 | 42.4 | 23.7 | 33.3 | 24.9 | 30.1 | 33.2 |
| | R@5k | 53.8 | 42.8 | 49.4 | 34.0 | 42.1 | 31.6 | 38.8 | 41.8 |
| ICT | R@2k | 41.6 | 35.9 | 43.9 | 27.8 | 34.0 | 24.5 | 27.8 | 33.6 |
| | R@5k | 52.5 | 46.1 | 51.6 | 39.0 | 42.8 | 32.9 | 38.5 | 43.3 |
| MSM | R@2k | 47.9 | 32.6 | 44.9 | 24.1 | 35.8 | 25.3 | 34.0 | 34.9 |
| | R@5k | 58.4 | 41.8 | 51.9 | 36.9 | 44.6 | 37.6 | 42.1 | 44.7 |

improvements on all the tasks, which demonstrates the effectiveness of our proposed MSM. (2) Under the setting of cross-lingual zero-shot transfer, it shows that our MSM outperforms other models by a large margin. Compared to strong XLM-R, MSM improves 7% MRR@100 on Mr. TyDi, 4.3% R@2k on XOR Retrieve, 1.8% mAP@20 of Mewsli-X, and 4.2% mAP@20 of LAReQA. (3) Under the setting of multi-lingual fine-tuning, the performance can be further improved by fine-tuning on multi-lingual train data and MSM can achieve the best results. However, there usually doesn't exist available multi-lingual data (such as Mewsli-X and LAReQA), especially for low-resource languages, and in this case MSM can achieve more gains for its stronger cross-lingual ability.

## 5.3 RESULTS ACROSS LANGUAGES

Table.2 show the results across languages on Mr. TyDi. The compared models are categorized into two kinds: mBERT, XLMR, CROP and MSM are unsupervised pre-trained models without any supervised data, while others are specialized multilingual encoders (Litschko et al., 2021) trained with parallel data or labeled data. DistilmBERT (Reimers & Gurevych, 2020) distills knowledge from the m-USE (Yang et al., 2019) teacher into a multilingual DistilBERT. InfoXLM (Chi et al., 2021) and LaBSE (Feng et al., 2022) encourage bilingual alignment with parallel data via ranking loss. They all use additional supervised data while our MSM only needs multi-lingual data. And in Appendix.A.4, we provide more comparison with these multilingual models. Through the detailed results in Tab.2, we demonstrate that MSM has consistent improvements over others on average

Table 4: Performance comparison of zero-shot cross-lingual retrieval when setting individual document encoder (and projection head) for English and other languages. *OTHS* means the average of languages except for *EN*.

| Method | Metrics | AR | BN | FI | ID | JA | KO | RU | SW | TE | TH | **EN** | **OTHS** | **AVG** |
|---|---|---|---|---|---|---|---|---|---|---|---|---|---|---|
| XLMR | MRR@100 | 30.8 | 30.2 | 23.5 | 33.5 | 23.5 | 26.6 | 25.7 | 24.0 | 26.6 | 36.3 | 27.1 | 28.1 | 28.0 |
| | Recall@100 | 72.3 | 78.4 | 67.0 | 79.8 | 66.5 | 64.7 | 65.0 | 57.2 | 72.7 | 84.6 | 74.3 | 70.8 | 71.1 |
| Share All | MRR@100 | 37.9 | 39.4 | 27.7 | 38.3 | 28.0 | 26.8 | 28.5 | 32.1 | 43.1 | 42.0 | 29.9 | 34.4 | **34.0** |
| | Recall@100 | 77.3 | 82.9 | 70.5 | 83.5 | 67.9 | 61.8 | 68.6 | 69.9 | 83.5 | 84.9 | 73.6 | 75.1 | **74.9** |
| Sep Doc | MRR@100 | 35.5 | 38.2 | 26.2 | 38.3 | 24.9 | 27.9 | 28.3 | 32.6 | 41.3 | 42.6 | 28.1 | 33.6 | 33.1 |
| | Recall@100 | 74.7 | 82.4 | 68.0 | 81.7 | 65.2 | 59.8 | 66.5 | 66.9 | 87.2 | 83.6 | 73.0 | 73.6 | 73.5 |
| Sep Doc + Head | MRR@100 | 35.8 | 39.7 | 28.8 | 36.5 | 24.7 | 25.1 | 26.9 | 32.0 | 20.8 | 38.8 | 28.0 | 30.9 | 30.6 |
| | Recall@100 | 75.4 | 76.1 | 70.2 | 81.7 | 63.5 | 59.5 | 65.6 | 69.1 | 68.2 | 82.2 | 72.4 | 71.2 | 71.3 |

results and most languages. LaBSE is slightly better on Recall@100 for it extends to a 500k vocabulary which is twice ours, and it utilizes parallel data. Interestingly though only fine-tuning on English data, MSM also achieves more gains in other languages. For example, MSM improves more than 5% on recall@100 on AR, FI, RU, and SW compared to XLMR, which clearly shows that MSM leads to better cross-lingual transfer compared to other baselines.

In Table.3, we mainly compare MSM with unsupervised pre-trained models. We reproduced two strong baselines proposed for monolingual retrieval, i.e. ICT and CROP, by extending them to multilingual corpora. We follow the original setting (Lee et al., 2019) for ICT and for CROP follow Wu et al. (2022a). It indicates that learning from two views of the same document (i.e. ICT and CROP) can achieve competitive results. Yet compared to them, MSM achieves more gains especially in low-resource languages, which indicates modeling sequential sentence relation across languages is indeed beneficial for cross-lingual transfer. More detailed results across different languages on other tasks can be seen in Appendix A.2 and there provides more analysis on multilinguality.

## 5.4 ANALYSIS OF CROSS-LINGUAL ABILITY

To investigate why MSM can advantage cross-lingual retrieval and how the cross-lingual ability emerges, we design some analytical experiments. Recall that in the zero-shot transfer setting, the pre-trained model is first fine-tuned on English data and then evaluated on all languages, which relies on the cross-lingual transfer ability from *en* to others. So in this experiment, we set individual the document encoder for *en* language and other languages to break the sentence relation shared between *en* and others, to see how it impacts the retrieval performance.

In Table.4, we report the results of different settings: Share All mean the original MSM where the document encoder is shared for all languages, Sep Doc sets two separate document encoder for *EN* and others, and Sep Doc + Head separates both encoder and projection head. The results clearly show that if *EN* and others don't share the same document encoder, the cross-lingual transfer ability drops rapidly. The previous works on multi-lingual BERTology (Conneau et al., 2020; Artetxe et al., 2020a; Rogers et al., 2020) found the shared model can learn the universal word embeddings across languages. Similar to it, our findings indicate that the shared document encoder benefits universal sentence embedding. This experiment further demonstrates that the sequential sentence relation is universal across languages, and modeling this relation is helpful for cross-lingual retrieval.

## 5.5 ABLATION STUDY

In this section, we conduct the ablation study on several components in our model. Considering computation efficiency and for a fair comparison, we fine-tune all the pre-trained models on NQ data and evaluate them on the target data.

**Ablation of Loss Function.** We first study the effectiveness of the hierarchical contrastive loss proposed in Eq.1. As shown in Table.5, $Cross\text{-}doc$ means only using cross-doc negatives without the intra-doc negatives. It results in poor performance due to the lack of utilization of intra-doc samples' information. When $w/o$ bias, it leads to a significant decrease for it regards intra-doc sentences as negatives, which would harm the representation space as we stated in Sec.3.2. We can

Table 5: Comparison of different settings for the hierarchical contrastive loss.

| Setting | Mr. TyDi | | XOR Retrieve | |
|---|---|---|---|---|
| | MRR@100 | R@100 | R@2k | R@5k |
| Cross-doc | 32.9 | 73.9 | 33.2 | 41.6 |
| $w/o$ bias $\alpha$ | 31.1 | 73.1 | 29.8 | 39.9 |
| $\mu = 0.3$ | 31.7 | 73.3 | 31.0 | 41.5 |
| $\mu = 0.5$ | **34.0** | **74.9** | **34.9** | **44.7** |
| $\mu = 0.7$ | 32.5 | 74.2 | 34.7 | 44.2 |

Table 6: Impact of contrastive negatives number. Best are marked bold.

| Size | Mr. TyDi | | XOR Retrieve | |
|---|---|---|---|---|
| | MRR@100 | R@100 | R@2k | R@5k |
| 256 | 31.4 | 73.6 | 33.2 | 42.9 |
| 512 | **34.0** | **74.9** | **34.9** | **44.7** |
| 1024 | 32.0 | 73.9 | 32.1 | 42.4 |
| 4096 | 31.8 | 72.8 | 32.3 | 43.2 |

Table 7: Impact of the projector settings. Best are marked bold.

| Setting | Mr. TyDi | | XOR Retrieve | |
|---|---|---|---|---|
| | MRR@100 | R@100 | R@2k | R@5k |
| $w/o$ PL | 27.0 | 66.1 | 30.3 | 39.1 |
| Shared PL | 32.1 | 71.0 | 34.5 | 43.8 |
| Asymmetric PL | **34.0** | **74.9** | **34.9** | **44.7** |

Table 8: Comparison of different document encoder layers. Best are marked bold.

| Layers | Mr. TyDi | | XOR Retrieve | |
|---|---|---|---|---|
| | MRR@100 | R@100 | R@2k | R@5k |
| 1 | 29.7 | 71.4 | 33.1 | 42.2 |
| 2 | **34.0** | **74.9** | **34.9** | **44.7** |
| 4 | 33.3 | 74.4 | 34.3 | 44.9 |
| 6 | 30.9 | 71.4 | 34.2 | 43.9 |

change the hyper $\mu$ to tune the impact of intra-doc negatives, and it gets the best results when setting $\mu$ at an appropriate value, which indicates ours can contribute to better representation learning.

**Impact of Contrastive Negative Size.** We analyze how the number of negatives influences performance and range it from 256 to 4096. As shown in Table.6, the performance increase as the negative size become larger and it has diminishing gain if the batch is sufficiently large. Interestingly the model performance does not improve when continually increasing batch size, which has been also observed in some work (Cao et al., 2022; Chen et al., 2021) on contrastive learning. In our work, it may be due to when the total negative number increases to a large number, the impact of intra-document negatives would be diminished and hurt the performance. By the way, the performance would be harmed by the instability when the batch size is too large (Chen et al., 2021).

**Impact of Projector Setting.** Existing work has shown the projection layers (Dong et al., 2022; Cao et al., 2022) between the representation and the contrastive loss affect the quality of learned representations. We explore the impact of different projection layer under different settings in Tab.7. Referring to Fig.1, Shared PL means the two projection layers share the same parameters, and Asymmetric PL means not sharing the layers. The results show that using an Asymmetric PL performs better than others and the removal of projection layers badly degrades the performance. One possible reason is that the projection layer can bridge the semantic gap between the representation of different samples in the embedding spaces.

**Impact of Decoder Layers.** We explore the impact of the size of document encoder layers in Table.8 and find a two-layer document encoder can achieve the best results. When the document encoder only has one layer, its capability is not enough to model sequential sentence relation, resulting in inefficient utilization of information. When the layer increase to larger, the masked sentence prediction task may depend more on the document encoder's ability and causes degradation of sentence representations, which is also in line with the findings of (Wu et al., 2022b; Lu et al., 2021).

## 6 CONCLUSION

In this paper, we propose a novel masked sentence model which leverages sequential sentence relation for pre-training to improve cross-lingual retrieval. It contains a two-level encoder in which the document encoder applied to the sentence vectors generated by the sentence encoder from a sequence of sentences. Then we propose a masked sentence prediction task to train the model, which masks and predicts the selected sentence vector via a hierarchical contrastive loss with sampled negatives. Through comprehensive experiments on 4 cross-lingual retrieval benchmark datasets, we demonstrate that MSM significantly outperforms existing advanced pre-training methods. Our further analysis and detailed ablation study clearly show the effectiveness and stronger cross-lingual retrieval capabilities of our approach.

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

# A  APPENDIX

## A.1  DETAILS OF DATASETS AND HYPERPARAMETERS

**CC-108.** Since Common Crawl (Wenzek et al., 2019) is not a public dataset, so we need to pre-process it by ourselves. Our reserved 108 languages are the union of the languages that XLMR and mBERT support. And we followed the processing method adopted by XLMR (Conneau et al., 2019) to pre-process CommonCrawl and retain 108 languages.

**Mr. TyDi.**  (Zhang et al., 2021b) It aims to evaluate cross-lingual passage retrieval with dense representations. Mr. TyDi is a multi-lingual benchmark dataset for mono-lingual query passage retrieval in eleven typologically distinct languages. Given a question in language $L$, the model should retrieve relevant texts in language $L$ that can respond to the query. As the original paper suggests, we take MRR@100 and Recall@100 for evaluation.

We mainly follow the open-source codebase DPR (Karpukhin et al., 2020) with minor modifications for multi-lingual models. When fine-tuned on MS MARCO in the zero-shot setting, we use AdamW optimizer with a learning rate of 2e-5. The model is trained for up to 3 epochs with a mini-batch size of 64. When fine-tuning on the NQ dataset, it is up to 40 epochs with a mini-batch size of 128. When further fine-tuning on Mr. TyDi's in-language data, it is 40 epochs, mini-batch size 128, and 1e-5 learning rate. Note all of them mainly follow what the original papers (Zhang et al., 2021b; 2022b) suggest. All these experiments are conducted on 8 NVIDIA Tesla A100 GPUs.

**XOR Retrieve.**  (Asai et al., 2021a) XOR QA is proposed for multilingual open-domain QA, and we take its sub-task XOR-Retrieve for our evaluation: given a question in $L$ (e.g., Korean), the task is to retrieve English passages that can answer the query. Following Asai et al. (2021a), we calculate the recall by computing the fraction of the questions for which the minimal answer is included in the top n tokens selected, and take R@2kt and R@5kt (kilo-tokens) as the metrics.

Similar to the settings of the previous one, we use AdamW Optimizer with a learning rate of 2e-5. The model is trained up to 40 epochs with a mini-batch size of 128 when fine-tuning on NQ dataset. And when further tuning on XOR's data, the hyper-parameter remains the same. We evaluate all the compared pre-trained models in the same pipelines on 8 NVIDIA Tesla A100 GPUs.

**Mewsli-X.**  (Ruder et al., 2021) Mewsli-X is built on top of Mewsli (Botha et al., 2020). We follow the setting of XTREME-R (Ruder et al., 2021), which builds Mewsli-X consisting of 15K mentions in 11 languages: given a mention in context, it is to retrieve the correct target entity description from a candidate pool ranging over 1M candidates across 50 languages.

We mainly follow the setting of XTREME-R (Hu et al., 2020). It adopts a 2e-5 learning rate, and it is trained for up to 2 epochs with batch size of 16. As the original paper suggests, all the evaluations are conducted on NVIDIA Tesla V100 GPU for a fair comparison.

**LAReQA.**  (Roy et al., 2020) Language Agnostic Retrieval Question Answering is a retrieval task in which each query has target answers in multiple languages, and models require retrieving all correct answers from the candidate pool regardless of language. Following Ruder et al. (2021), we use the LAReQA XQuAD-R dataset which consists of 13,090 questions, each of which has 11 target answers (in 11 different languages) within 13,014 candidate answer sentences.

Following Ruder et al. (2021), we use the LAReQA XQuAD-R dataset which consists of 13,090 questions, each of which has 11 target answers (in 11 different languages) within 13,014 candidate answer sentences. It also follows the setting proposed by XTREME-R (Hu et al., 2020). It adopts a 2e-5 learning rate and it is trained up to 3 epochs with batch size of 4. All the evaluations are conducted on NVIDIA Tesla V100 GPU.

## A.2  DETAILED RESULTS ACROSS LANGUAGES

We show the detailed results for each task across different languages corresponding to Table.1. Specifically, the results of Mr. TyDi are as shown in Table.9, XOR Retrieve in Table.10, Mewsli-X in Table.11, and LAReQA in Table.12.

Through the detailed results across languages, there are some findings on multilinguality: (1) MSM can achieve more gains in low-resource language. For example, in zero-shot setting of Mr. TyDi

Table 9: Mr. TyDi results across languages. We report MRR@100 and Recall@100 on the test sets of Mr. TyDi in the two settings.

| Method | Metrics | AR | BN | EN | FI | ID | JA | KO | RU | SW | TE | TH | AVG |
|---|---|---|---|---|---|---|---|---|---|---|---|---|---|
| *Cross-lingual zero-shot transfer (models are fine-tuned on English data)* | | | | | | | | | | | | | |
| mBERT | MRR@100 | 45.3 | 38.8 | 29.1 | 28.7 | 35.5 | 29.6 | 29.8 | 32.6 | 27.8 | 40.8 | 25.3 | 33.0 |
|  | Recall@100 | 77.8 | 84.7 | 73.8 | 65.8 | 72.9 | 68.4 | 59.9 | 71.1 | 56.4 | 76.8 | 59.2 | 69.7 |
| XLMR | MRR@100 | 43.8 | 41.2 | 29.3 | 33.2 | 45.4 | 27.0 | 33.4 | 32.2 | 35.3 | 44.5 | 49.7 | 37.7 |
|  | Recall@100 | 77.4 | 81.5 | 68.5 | 72.2 | 81.8 | 61.2 | 66.5 | 66.4 | 63.9 | 75.5 | 85.4 | 72.8 |
| MSM | MRR@100 | 51.6 | 53.0 | 31.6 | 39.4 | 50.5 | 32.0 | 36.8 | 37.2 | 43.4 | 62.6 | 53.5 | 44.7 |
|  | Recall@100 | 83.0 | 83.8 | 73.9 | 77.9 | 85.7 | 67.5 | 70.3 | 71.4 | 73.0 | 89.8 | 88.2 | 78.6 |
| *Multi-lingual fine-tune (models are fine-tuned on the multi-lingual data)* | | | | | | | | | | | | | |
| mBERT | MRR@100 | 66.8 | 63.3 | 50.8 | 55.0 | 55.7 | 47.8 | 43.3 | 47.3 | 60.5 | 85.8 | 57.7 | 57.6 |
|  | Recall@100 | 90.3 | 95.0 | 89.0 | 85.8 | 89.3 | 81.6 | 80.2 | 85.6 | 86.7 | 96.3 | 87.2 | 87.9 |
| XLMR | MRR@100 | 66.4 | 66.0 | 48.5 | 53.7 | 58.3 | 45.4 | 44.3 | 47.4 | 61.4 | 86.2 | 65.7 | 58.5 |
|  | Recall@100 | 89.4 | 93.2 | 84.3 | 86.6 | 90.0 | 82.3 | 79.3 | 82.8 | 86.4 | 97.4 | 93.8 | 87.8 |
| MSM | MRR@100 | 67.7 | 69.9 | 49.6 | 55.1 | 61.2 | 48.2 | 49.4 | 47.5 | 63.9 | 85.2 | 67.7 | 60.5 |
|  | Recall@100 | 90.5 | 95.9 | 86.5 | 88.0 | 90.1 | 82.0 | 81.6 | 82.5 | 88.9 | 97.6 | 95.0 | 89.0 |

Table 10: XOR Retrieve results across languages. We report R@2k and R@5 metrics on the test sets in the two settings.

| Method | Metrics | AR | BN | FI | JA | KO | RU | TE | AVG |
|---|---|---|---|---|---|---|---|---|---|
| *Cross-lingual zero-shot transfer (models are fine-tuned on English data)* | | | | | | | | | |
| mBERT | R@2k | 31.1 | 26.6 | 38.5 | 32.4 | 38.6 | 24.9 | 29.1 | 31.6 |
|  | R@5k | 44.1 | 36.2 | 48.1 | 41.5 | 48.1 | 38.4 | 39.5 | 42.3 |
| XLMR | R@2k | 39.9 | 25.7 | 41.1 | 27.8 | 31.9 | 22.4 | 25.2 | 30.6 |
|  | R@5k | 49.6 | 34.9 | 50.0 | 34.9 | 41.8 | 30.4 | 34.0 | 39.3 |
| MSM | R@2k | 47.9 | 32.6 | 44.9 | 24.1 | 35.8 | 25.3 | 34.0 | 34.9 |
|  | R@5k | 58.4 | 41.8 | 51.9 | 36.9 | 44.6 | 37.6 | 42.1 | 44.7 |
| *Multi-lingual fine-tune (models are fine-tuned on the multi-lingual data)* | | | | | | | | | |
| mBERT | R@2k | 51.7 | 52.7 | 52.2 | 41.9 | 51.9 | 47.5 | 40.5 | 48.3 |
|  | R@5k | 58.4 | 61.5 | 58.6 | 50.6 | 63.2 | 54.9 | 51.8 | 57.0 |
| XLMR | R@2k | 59.2 | 48.7 | 46.8 | 36.1 | 46.0 | 35.4 | 43.4 | 45.1 |
|  | R@5k | 67.2 | 58.6 | 53.5 | 45.6 | 56.1 | 45.1 | 51.5 | 53.9 |
| MSM | R@2k | 61.8 | 55.3 | 51.3 | 36.9 | 50.5 | 41.4 | 43.0 | 48.6 |
|  | R@5k | 68.9 | 63.8 | 59.9 | 48.1 | 56.1 | 52.7 | 51.5 | 57.3 |

(Tab.9) it improves + 8.1 MRR@100 for SW and + 11.8 for BN compared to XLMR, and for XOR Retrieve (Tab.10) + 8.8 R@5k for TE. Though there are limited data in low-resource languages and the model suffers from the curse of multilinguality, our MSM can lead to better transfer to them, benefiting from modeling the sequential sentence relation across languages. (2) The target languages closer to pivot language (i.e. EN in our experiment) usually perform better and achieve more improvements. On Mewsli-X task (Tab.11) MSM can improve + 5.3 MAP for language DE while only + 1.3 for UK, for German (DE) is more similar to English in both scripts and language family (Ruder et al., 2021). Similar observations also exist in LAReQA (Tab.12) that MSM performs better and improves more on DE, EL, ES and poorer on ZH and TH. (3) The multi-lingual data lead to better cross-lingual retrieval performance. It can be seen in Tab.9 and Tab.10, the performance can be further improved after fine-tuning multi-lingual train data. It indicates that the target languages can benefit from the other languages' data, and also shows that fine-tuning on multi-lingual data is necessary if available. It is worth mentioning that in this setting MSM can also achieve more gains, which demonstrates better cross-lingual transfer ability.

Table 11: Mewsli-X results across different input languages. We report the mean average precision@20 (mAP@20) results.

| Method | AR | DE | EN | ES | FA | JA | PL | RO | TA | TR | UK | AVG |
|---|---|---|---|---|---|---|---|---|---|---|---|---|
| *Cross-lingual zero-shot transfer (models are fine-tuned on English data)* | | | | | | | | | | | | |
| mBERT | 14.2 | 65.5 | 57.4 | 55.7 | 11.3 | 44.8 | 57.2 | 38.2 | 5.8 | 42.7 | 36.2 | 39.0 |
| XLMR | 15.5 | 62.7 | 57.2 | 53.5 | 12.5 | 46.2 | 59.3 | 34.1 | 7.5 | 51.8 | 36.0 | 39.7 |
| MSM | 18.6 | 68.0 | 58.7 | 57.3 | 13.7 | 46.3 | 60.2 | 36.3 | 7.6 | 52.8 | 37.3 | **41.5** |

Table 12: LAReQA results across different question languages. We report the mean average precision@20 (mAP@20) results.

| Method | AR | DE | EL | EN | ES | HI | RU | TH | TR | VI | ZH | AVG |
|---|---|---|---|---|---|---|---|---|---|---|---|---|
| *Cross-lingual zero-shot transfer (models are fine-tuned on English data)* | | | | | | | | | | | | |
| mBERT | 20.1 | 32.2 | 19.9 | 34.8 | 33.5 | 15.3 | 29.6 | 7.2 | 23.3 | 27.6 | 26.2 | 24.5 |
| XLMR | 21.4 | 31.7 | 26.7 | 36.7 | 34.3 | 25.2 | 32.3 | 27.8 | 29.0 | 29.1 | 28.5 | 29.3 |
| MSM | 28.0 | 36.1 | 31.9 | 40.6 | 38.3 | 29.4 | 36.0 | 30.2 | 34.3 | 33.2 | 29.7 | **33.4** |

## A.3 COMPARISON WITH SEVERAL EXISTING METHODS

Table 13 shows the comparison of MSM and several existing retrieval approaches. DistilmBERT (Reimers & Gurevych, 2020) distills knowledge from m-USE (Yang et al., 2019) trained on labeled pair data into mBERT. LaBSE (Feng et al., 2022) and InfoXLM (Chi et al., 2021) encourage bilingual alignment via a translation ranking loss, and also trained with MLM and TLM tasks (Lample & Conneau, 2019). InfoXLM adopts the momentum contrast and LaBSE proposed additive margin softmax for contrastive learning. They all use additional parallel corpora, while our MSM only needs multi-lingual data without relying on any parallel or labeled data.

Among unsupervised pre-trained models, mBERT (Devlin et al., 2019) and XLMR (Conneau et al., 2019) are general-purpose multilingual text encoders trained with MLM. XLMR-Long (Sagen, 2021) (or XLMR Longformer) is an XLMR model that has been extended to allow sequence lengths up to 4096 tokens. mContriever (Izacard et al., 2021b) and CCP (Wu et al., 2022a) similarly mine positive pairs by cropping two spans in a document. The former proposes random cropping while the latter utilizes two sentences. However, the quality of the cropped pairs is not guaranteed. In contrast, MSM utilizes a sequence of sentences in a document and models the universal sentence relation across languages via an explicit document encoder, which results in better cross-lingual retrieval capability.

Table 13: Comparison with existing approaches. For the training objective, BI means bi-lingual pair alignment task, and CROP means contrastive learning with cropped spans. For the corpora, CC means CommonCrawl, mWiki means multi-lingual data from Wikipedia, and Bi-lingual may include MultiUN, OPUS, WikiMatrix, etc (Chi et al., 2021), which depends on models.

| Model | #lg | Objective Function | Corpora |
|---|---|---|---|
| DistilmBERT (Reimers & Gurevych, 2020) | 53 | Distillation | Bi-lingual |
| LaBSE (Feng et al., 2022) | 109 | MLM + TLM + BI | CC + mWiki + Bi-lingual |
| InfoXLM (Chi et al., 2021) | 94 | MLM + TLM + BI | CC + Bi-lingual |
| mBERT (Devlin et al., 2019) | 104 | MLM | mWiki |
| XLMR (Conneau et al., 2019) | 100 | MLM | CC |
| XLMR-Long (Sagen, 2021) | 100 | MLM | CC + Wiki |
| mContriever (Izacard et al., 2021b) | 29 | CROP | CC + mWiki |
| CCP (Wu et al., 2022a) | 108 | MLM + CROP | CC |
| MSM | 108 | MLM + MSM | CC |

Table 14: Cross-lingual zero-shot transfer retrieval performance for different size models. We report MRR@100 and Recall@100 on the test sets of Mr. TyDi after finetuning on NQ dataset.

| Method | Metrics | AR | BN | EN | FI | ID | JA | KO | RU | SW | TE | TH | AVG |
|--------|---------|----|----|----|----|----|----|----|----|----|----|----|-----|
| mBERT | MRR@100 | 28.7 | 33.4 | 28.1 | 24.0 | 32.0 | 24.3 | 21.8 | 29.0 | 18.7 | 14.5 | 19.3 | 24.9 |
|  | Recall@100 | 68.3 | 79.3 | 76.1 | 65.0 | 74.1 | 65.3 | 57.9 | 69.2 | 51.7 | 46.1 | 54.0 | 64.3 |
| XLMR Base | MRR@100 | 30.8 | 30.2 | 27.1 | 23.5 | 33.5 | 23.5 | 26.6 | 25.7 | 24.0 | 26.6 | 36.3 | 28.0 |
|  | Recall@100 | 72.3 | 78.4 | 74.3 | 67.0 | 79.8 | 66.5 | 64.7 | 65.0 | 57.2 | 72.7 | 84.6 | 71.1 |
| XLMR Large | MRR@100 | 36.5 | 37.4 | 27.5 | 31.8 | 39.5 | 29.9 | 30.4 | 30.6 | 27.4 | 34.6 | 40.1 | 33.3 |
|  | Recall@100 | 81.3 | 84.2 | 77.6 | 78.2 | 88.6 | 78.5 | 72.7 | 77.4 | 63.3 | 87.5 | 88.2 | 79.8 |
| InfoXLM Large | MRR@100 | 37.2 | 50.4 | 31.4 | 30.9 | 37.6 | 27.1 | 30.9 | 32.5 | 39.4 | 46.5 | 37.4 | 36.5 |
|  | Recall@100 | 76.2 | 91.0 | 78.3 | 76.0 | 85.2 | 66.9 | 64.4 | 74.4 | 75.0 | 88.9 | 83.4 | 78.2 |
| CCP Large | MRR@100 | 42.6 | 45.7 | 35.9 | 37.2 | 46.2 | 37.7 | 34.6 | 36.0 | 39.2 | 47.0 | 48.9 | 41.0 |
|  | Recall@100 | 82.0 | 88.3 | 80.1 | 78.7 | 87.5 | 80.0 | 73.2 | 77.2 | 75.1 | 88.8 | 88.9 | 81.8 |
| MSM Base | MRR@100 | 37.9 | 39.4 | 29.9 | 27.7 | 38.3 | 28.0 | 26.8 | 28.5 | 32.1 | 43.1 | 42.0 | 34.0 |
|  | Recall@100 | 77.3 | 82.9 | 73.6 | 70.5 | 83.5 | 67.9 | 61.8 | 68.6 | 69.9 | 83.5 | 84.9 | 74.9 |
| MSM Large | MRR@100 | 45.6 | 55.7 | 33.8 | 39.1 | 46.8 | 40.3 | 38.8 | 37.9 | 41.8 | 46.5 | 50.8 | 43.4 |
|  | Recall@100 | 83.3 | 90.1 | 78.5 | 79.9 | 85.9 | 78.1 | 73.1 | 77.1 | 76.4 | 86.9 | 88.9 | 81.6 |

## A.4 Performance of different model size

Motivated by the recent progress of giant models, we also increase the model capability. Considering expensive computation, it is initialized with XLMR-large and other settings keep the same as the base model. As shown in Tab.14, we report the zero-shot transfer results on Mr. TyDi after fine-tuning on NQ data. It clearly shows that as the model capacity increase, the performance on the downstream task can be consistently improved. We also report several strong large-size pre-trained models, including InfoXLM and CCP which are both initialized with XLM-R Large. Compared to them, MSM-Large achieves outperforming results on MRR@100 and comparable results on Recall@100, which further demonstrates the effectiveness of MSM in different model capabilities.

## A.5 Comparison to MT-Based Cross-lingual Retrieval

In this section, we compare the model performance to the MT-based (machine translation based) cross-lingual retrieval. As shown in Table.15, we provide four MT-based baselines borrowed from (Asai et al., 2021a), all of which first translate the query to the document language using a translation system and then perform monolingual retrieval. For translation systems, *GMT* means Google's online machine translation service and *White-MT* means white-box translation model based on autoregressive transformers. For the monolingual retrieval model, *PATH* means Path Retriever (Asai et al., 2019), a graph-based recurrent retrieval approach, and DPR (Karpukhin et al., 2020) is a typical bi-encoder based retriever. Both of them are trained on the human translated queries with the annotated gold paragraph data of XOR Retrieve.

Results in Table.15 clearly shows that MSM can outperform While-box MT-based methods by a large margin, which demonstrates the effectiveness of MSM. Moreover, upgrading the MT system to GMT achieve even better results, due to the superiority of industrial MT systems (large parallel data, models and pipelines, etc.) (Asai et al., 2021a). These observations indicate that the performance of MT-base methods heavily depends on the quality of MT system. However, GMT is a black-box system so it's difficult to be analyzed. By the way, the MT-based method relies on the two-stage pipeline that first translates and then retrieves, which may lead to cumulative errors. In contrast, our MSM is a universal pre-trained model and can be easily applied in bi-encoder retrievers, which have more advantages in terms of deployments and diagnosis.

## A.6 Monolingual Experiment

We adapt our MSM pre-training method to the monolingual domain, in which we narrow the train corpus and model to English only. We initialize the sentence encoder with ERNIE-2.0-Base model and others adopt the same setting to our multi-lingual experiment. In Table.16, we report the performance on the Natural Question (Kwiatkowski et al., 2019) test set after fine-tuning. It shows that our proposed unsupervised model achieves better performance than advanced baselines including

Table 15: Performance comparison to MT-Based Cross-lingual Retrieval on XOR Retrieve across languages. We report R@2k and R@5k results.

| Method | Metrics | AR | BN | FI | JA | KO | RU | TE | AVG |
|---|---|---|---|---|---|---|---|---|---|
| GMT + PATH | R@2k | 59.1 | 58.2 | 60.3 | 50.0 | 50.3 | 54.1 | 58.0 | 58.2 |
| | R@5k | 63.3 | 78.9 | 64.1 | 52.3 | 54.0 | 56.5 | 62.5 | 61.7 |
| GMT + DPR | R@2k | 61.7 | 72.0 | 60.6 | 52.1 | 57.9 | 51.2 | 59.4 | 59.3 |
| | R@5k | 67.5 | 83.2 | 68.1 | 60.1 | 66.3 | 60.4 | 65.0 | 67.2 |
| White-MT + PATH | R@2k | 45.0 | 60.9 | 56.6 | 36.7 | 33.8 | 34.7 | 15.7 | 40.5 |
| | R@5k | 51.6 | 64.8 | 59.5 | 41.7 | 37.6 | 38.1 | 18.1 | 44.5 |
| White-MT + DPR | R@2k | 48.3 | 54.4 | 56.7 | 41.8 | 39.4 | 39.6 | 18.7 | 42.7 |
| | R@5k | 52.5 | 63.2 | 65.9 | 52.1 | 46.5 | 47.3 | 22.7 | 50.0 |
| XLM-R | R@2k | 59.2 | 48.7 | 46.8 | 36.1 | 46.0 | 35.4 | 43.4 | 45.1 |
| | R@5k | 67.2 | 58.6 | 53.5 | 45.6 | 56.1 | 45.1 | 51.5 | 53.9 |
| MSM | R@2k | 61.8 | 55.3 | 51.3 | 36.9 | 50.5 | 41.4 | 43.0 | 48.6 |
| | R@5k | 68.9 | 63.8 | 59.9 | 48.1 | 56.1 | 52.7 | 51.5 | 57.3 |

Table 16: Monolingual Retrieval performance on Natural Question test. The best performing models are marked bold and the results unavailable are left blank.

| Method | R@5 | R@20 | R@100 |
|---|---|---|---|
| BM25 (Yang et al., 2017) | - | 59.1 | 73.7 |
| BERT (Karpukhin et al., 2020) | - | 78.4 | 85.3 |
| RoBERTa (Liu et al., 2019) | 64.9 | 76.8 | 84.7 |
| ERNIE2.0 (Sun et al., 2020) | 68.7 | 79.8 | 86.1 |
| MSM (ours) | **69.3** | **80.3** | **86.5** |

BERT and ERNIE2.0 (Sun et al., 2020). It further proves our MSM is a general pre-training method tailored for dense retrieval including monolingual and multi-lingual domains.

