# OpenReview forum: "Modeling Sequential Sentence Relation to Improve Cross-lingual Dense Retrieval"
_ICLR.cc/2023/Conference — ICLR 2023 poster_

### Official Review · Reviewer_Hf6d · 2022-10-21

**Confidence:** 4
**Correctness:** 3
**Technical Novelty And Significance:** 3
**Empirical Novelty And Significance:** 3
**Recommendation:** 6

**Clarity, Quality, Novelty And Reproducibility:**

Clarity & Quality - The paper is clear and well-explained.
Novelty - The paper isn't particularly novel, but it combines well-established InfoNCE-like contrastive learning approaches with multilingual pre-trained masked language models.
Reproducibility - The approach is simple enough that it should be possible to implement without too many details. The paper however does lack some information regarding the number of negatives used or dataset processing details.

**Strength And Weaknesses:**

Strengths
1. The paper is well-written and explained.
2. The motivation for the overall approach is clear and the idea is conceptually simple. With only a few different moving parts in the model, the authors are to run fairly thorough ablations.
3. The results on both multilingual and cross-lingual retrieval benchmarks are significantly better than the baselines considered.

Weaknesses
1.  The paper's motivation seems to focus a little too much on the "cross-lingual" aspect of things rather than just as a multilingual hierarchical contrastive learning approach. The authors' arguments center around the fact that sentences are ordered similarly with documents across different languages.
2. The paper lacks some details about the dataset used. The paper cites the CC-100 paper, but was it exactly CC-100? Or did you pre-process commoncrawl yourselves?


**Summary Of The Paper:**

The paper presents a contrastive learning approach to learning hierarchical document representations for multilingual and cross-lingual retrieval. The approach encodes a document in a hierarchical fashion by encoding each sentence in a document into a single fixed-length vector with an XLM-R encoder. The sequence of sentence representations is encoded by a document encoder that is trained from scratch. The overall training objective is a combination of a contrastive and masked language modeling loss. The contrastive loss masks out one of the sentence representations that is provided to the document encoder and uses an InfoNCE-like objective with intra-document as well as cross-document negatives.

The model is trained on CC-100-like data on 108 languages for 200k steps and evaluated on multilingual and cross-lingual retrieval benchmarks. The model is able to do significantly better when compared with mBERT, XLM-R and ICT baselines.

**Summary Of The Review:**

A simple and easy-to-implement contrastive learning approach to improve multilingual and cross-lingual retrieval. The baselines aren't particularly strong, but still a strong contribution.

---

> ### Author Response · Authors · 2022-11-19
> **Response to Reviewer Hf6d**
>
> > Q1. The paper's motivation seems to focus a little too much on the "cross-lingual" aspect of things rather than just as a multilingual hierarchical contrastive learning approach. The authors' arguments center around the fact that sentences are ordered similarly with documents across different languages.
>
> Thanks for your insightful comments. Indeed, the two aspects, namely the motivation on cross-lingual aspect and the multilingual hierarchical approach, are highly correlated rather than independent from each other. We start from the idea that sentences are ordered similarly with documents across different languages. And to model this relation, we design the hierarchical learning approach including the hierarchical model and masked sentence prediction task. In turn, our experiments demonstrate that the hierarchical learning approach is crucial for stronger cross-lingual retrieval capabilities, thus resulting in better downstream cross-lingual performance.
>
>
>
> > Q2. The paper lacks some details about the dataset used. The paper cites the CC-100 paper, but was it exactly CC-100? Or did you pre-process commoncrawl yourselves?
>
> Yes, since Common Crawl is not a public dataset, so we need to pre-process it  by ourselves. Our reserved 108 languages are the union of the languages that XLMR and mBERT support, which covers more languages for larger application scope. And we followed the public processing code [1] adopted by XLMR to pre-process CommonCrawl [2] and retain 108 languages for our pre-training.
>
>
>
> Thanks for your constructive feedback, and we have revised the paper to make them more clear. Please let me know if you have any other questions.
>
> References:
>
> [1] https://github.com/facebookresearch/cc_net
>
> [2] Wenzek et al. LREC 2020. https://aclanthology.org/2020.lrec-1.494/

---

### Official Review · Reviewer_TsJC · 2022-10-24

**Confidence:** 4
**Clarity, Quality, Novelty And Reproducibility:** Good.
**Correctness:** 3
**Technical Novelty And Significance:** 3
**Empirical Novelty And Significance:** 2
**Recommendation:** 6

**Strength And Weaknesses:**

Strength:
- The paper is well organized and easy to follow.
- The proposed hierarchical contrastive learning pre-training objective is simple and effective on cross-lingual retrieval tasks.

Weaknesses:
- I am a bit surprised that the document encoder is discarded after pre-training. I thought self-attention in the document encoder will be important for modeling the universal sequential sentence relation across languages.  It is unclear to me that why only sentence encoder is used for downstream finetuning.

**Summary Of The Paper:**

This paper introduces MSM, a hierarchical multilingual encoder pre-trained with hierarchical contrastive learning. MSM applies XLMR as sentence encoder, and uses another transformer encoder to encode the document level context. During training, both masked language model loss and masked sentence model loss are applied. At inference time, the document encoder is discarded and only the sentence encoder is used. Experimental results showed that the hierarchical pre-training strategy yielded SOTA results on four cross-lingual retrieval tasks.

**Summary Of The Review:**

The paper introduces a simple yet effective method for improving cross-lingual retrieval. Extensive experiments have been conducted to demonstrate the effectiveness of the proposed method.

---

> ### Author Response · Authors · 2022-11-19
> **Response to Reviewer TsJC**
>
> > Q1. Why the document encoder is discarded after pre-training ?
>
> Thanks for your valuable comments. There are some points about this question:
>
> a). In our hierarchical approach, the document encoder plays as a bottleneck: the sentence encoder press the sentence semantics into sentence vectors, and the document encoder leverage the limited information to predict the masked vector, thus enforcing an information bottleneck on the sentence encoder for better representations. There are several papers also utilizing the similar method, such as Condenser [1] and RetroMAE [2].
>
> b). According to our analysis in Sec.5.4, the shared document encoder helps to model the sequential sentence relation across languages. So it can promote sentence representations for better cross-lingual ability, which is more vital for downstream retrieval tasks. It is similar to some findings on mBERTology [3, 4] which find the shared model can learn the universal word embeddings across languages. And our experiments indicate that shared document encoder does benefit universal sentence embedding.
>
> c). By the way, the sentence encoder has the same architecture as XLMR, which ensures a fair comparison.
>
>
>
> We have updated the paper to make these points more clearly. Please let me know if you have any other questions.
>
>
>
> References:
>
> [1] Gao et al. EMNLP 2021. https://aclanthology.org/2021.emnlp-main.75/
>
> [2] Xiao et al. EMNLP 2022. https://arxiv.org/abs/2205.12035
>
> [3] Doddapaneni et al. 2021 https://arxiv.org/abs/2107.00676
>
> [4] Rogers et al. TACL 2020. https://arxiv.org/abs/2002.12327

---

### Official Review · Reviewer_VGZA · 2022-10-25

**Confidence:** 3
**Clarity, Quality, Novelty And Reproducibility:** This paper is easy to read and can be…
**Correctness:** 3
**Technical Novelty And Significance:** 3
**Empirical Novelty And Significance:** 3
**Recommendation:** 8

**Strength And Weaknesses:**

Overall this paper represents a great result. The idea of using sequential sentence relation to facilitate cross-lingual representation learning is both novel and effective. The bias term added to combine intra and inter-document negatives is novel.

One concern I have for the proposed approach is its efficiency -- masking document representation with leave-one-sentence-out scheme seems very expensive, since the document encoder needs to be recomputed for every masking.

It is also counter intuitive that more negatives performs worse than 512 negatives -- usually larger patch sizes improves contrastive learning.


**Summary Of The Paper:**

This work aims to improve the cross-lingual retrieval capability of dual encoder models.
Given access to multi-lingual document corpus and the intuition that sentence ordering across languages are often similar, this work proposes to use sequential sentence relation to facilitate cross-lingual representation learning.

More specifically it proposes a hierarchical architecture (MSM), which first encodes sentences to embeddings, and then performs doc level MLM (with contrastive loss) to model sentence relations. The MLM task is constructed by pairing masked sentence representation produced by the doc transformer and the original sentence embedding. In order to combine intra and inter document negative examples, an adaptive bias term is added to the logits to make sure that one does not dominate the other.

The model is initialized from XLMR, and the pretraining loss combines token MLM and sentence MLM. The pretraining corpus is a clean version of common crawl (similar to that of XMLR).

Experiments on 4 cross lingual retrieval tasks show that MSM significantly improves cross-lingual transferability over baselines (XLMR, mBERT), and also improves the fine-tune settings.
Experiment against stronger baselines (ICT, MSM) on 2 of the tasks shows improvement of MSM over baselines. These stronger baselines are previously only used in monolingual settings.
Oblation study shows that the best setting is 1) combining inter and intra-document negatives 2) 512 negatives 3) asymmetric projection layer  4) 2 layers of document encoder


**Summary Of The Review:**

Novel and effective approach, with a few minor questions.

---

> ### Author Response · Authors · 2022-11-19
> **Response to Reviewer VGZA**
>
> > Q1. One concern I have for the proposed approach is its efficiency -- masking document representation with leave-one-sentence-out scheme seems very expensive, since the document encoder needs to be recomputed for every masking.
>
> Thanks for your insightful comments. Indeed our masked sentence scheme is so efficient. First, the sentences in a document pass through the sentence encoder only once without recomputation. As for the document encoder, the length of document encoder's input is not long, for the number of sentences in a document is not long. And our document encoder is also shallow. These factors make our approach very efficient without much computation.
>
>
>
> > Q2. It is also counter intuitive that more negatives performs worse than 512 negatives -- usually larger batch sizes improves contrastive learning.
>
> Thanks for your valuable comments.
>
> a). Recall that our negatives contain a limited number of Intra-doc Negatives (for the sentence in a document is limited) and a varying number of Cross-doc Negatives. And the results in Table.5 show Intra-doc Negatives play an important role in our hierarchical contrastive loss. So when the total negative number increases to a large number, the impact of intra-document negatives would be diminished and hurt the performance.
>
> b). Recent works [1, 2] also revealed that keep increasing the batch size did not necessarily bring improvement, which is also in line with our experiment. It has diminishing gains if the batch is sufficiently large, and the performance would be harmed by the instability when the batch size is too large [1].
>
>
>
> We have updated the paper to make these points more clearly. Please let me know if anything is unclear or if you have any additional concerns.
>
>
>
> [1] Xinlei Chen et al. ICCV 2021. https://arxiv.org/pdf/2104.02057.pdf
>
> [2] Rui Cao et al. ACL 2022. https://arxiv.org/abs/2202.13093

---

> > ### Comment · Reviewer_VGZA · 2022-11-20
> > **I am satisfied with the answers to my questions**
> >
> > 4096 in Xinlei Chen et al. ICCV 2021 is much larger than the optimal 512 here, but I understand that there is the issue of the ratio of intra doc negatives.

---

### Official Review · Reviewer_XpJe · 2022-10-27

**Confidence:** 5
**Correctness:** 2
**Technical Novelty And Significance:** 2
**Empirical Novelty And Significance:** 2
**Recommendation:** 3

**Clarity, Quality, Novelty And Reproducibility:**

The paper will not have any major impact as it omits many major baselines and a lot of very relevant work, offering only basic comparisons and lacking insightful side analyses. It makes a minor methodological contribution by combining document-level and sentence-level masked language modeling, which is not evaluated against cutting-edge CLIR methods.

It should be possible to reproduce the main results in the paper - it does not mention whether the results are average over several random seeds or not (and which random seed was used).

**Strength And Weaknesses:**

Strengths:
- The idea of MSM is simple and neat, although it is very similar to the idea of next sentence prediction - the difference here is that the model performs masked sentence prediction.
- The ablation study shows the usefulness of introducing the MSM objective.

The paper is simply not at a state to be considered for publication, with a series of major flaws as follows.
Weaknesses:
- Only a partial awareness of very related work on cross-lingual information retrieval (CLIR), with many strong reference works and baselines omitted from the paper completely and omitted from the comparisons. Here, only a few directly relevant papers are mentioned:
-- https://arxiv.org/pdf/2112.09118.pdf
-- https://arxiv.org/abs/2204.02292
-- https://arxiv.org/abs/2004.09813
-- https://arxiv.org/pdf/2101.08370.pdf
-- There is also work on multilingual Longformers (e.g., https://huggingface.co/markussagen/xlm-roberta-longformer-base-4096)
- Related to the point above, those papers provide much stronger baselines - the baselines in this submission are simply weak and inadequate. When doing sentence retrieval, the paper should compare against strong multilingual sentence encoders and not the original off-the-shelf models.
- The paper also does conflate query-passage and sentence retrieval, and does not evaluate on document retrieval at all. There are huge differences on how to approach each 'granularity of information' when doing retrieval, and the paper does not seem to pay attention to that: e.g., check this work for further details: https://arxiv.org/pdf/2101.08370.pdf
- The paper also critically requires parallel data to work -> if one has parallel data, one of the must-have baselines are also MT-based query-translate or passage-translate approaches which sometimes/often work better than standard encoder-based approaches.
- There are no discussions on how different target languages might impact the results: are all the languages equally difficult, which ones might cause major drops of performance and, most importantly, why? The paper treats multilinguality very superficially.

There are other (minor) weaknesses, including problems with language and presentation, but the major ones are mostly listed above.


**Summary Of The Paper:**

This work proposes a new method for cross-lingual and multilingual dense retrieval, focusing on applications of query-passage and sentence retrieval in multilingual setups. The approach relies on parallel document-aligned and assumes that the sentences in such documents are roughly in the same order; the idea is then to combine the standard MLM objective with its bilingual sentence-level variant, termed Masked Sentence Modeling (MSM), hoping to guess the correct sentence (once the sentence is masked in the document), where the document encoder is shared across languages.

The experiments are conducted on four retrieval tasks from prior work, and gains over (mostly) weak baselines are reported.

**Summary Of The Review:**

The paper lacks strong baselines, shows only partial awareness of the current cutting-edge CLIR methodology. and it is difficult to contextualise its results (does it really bring any major improvement for CLIR?). It also does not delve deeper into intricacies of multilinguality and differences between sentence/passage/document retrieval. There are also presentation problems which make the paper seem incomplete and half-finished

---

> ### Author Response · Authors · 2022-11-19
> **Response to Reviewer XpJe, Part 1**
>
> Thanks for your insightful and invaluable comments. Our paper has been carefully revised according to the comments and the revised version has been submitted to OpenReview. Below is our point-to-point response.
>
> > Q1. Only a partial awareness of very related work on cross-lingual information retrieval (CLIR), with many strong reference works and baselines omitted from the paper completely.
> >
> > Q2.  Related to the point above, those papers provide much stronger baselines - the baselines in this submission are simply weak and inadequate.
>
> Thanks for the valuable comments. First we add more strong baselines, and then we clarify some misunderstandings and the differences between these methods and our MSM.
>
> The below table shows the performance comparison on Mr. TyDi after fine-tuning on MS MARCO.
>
> | Method      | Metrics    | AR   | BN   | EN   | FI   | ID   | JA   | KO   | RU   | SW   | TE   | TH   | AVG  |
> | :---------- | ---------- | ---- | ---- | ---- | ---- | ---- | ---- | ---- | ---- | ---- | ---- | ---- | ---- |
> | DistilmBERT | MRR@100    | 40.8 | -    | 29.9 | 26.7 | 39.7 | 27.0 | 32.2 | 29.4 | 22.0 | -    | 26.5 | 30.5 |
> |             | Recall@100 | 79.7 | -    | 71.0 | 64.1 | 79.7 | 65.0 | 64.4 | 62.6 | 48.2 | -    | 60.9 | 66.2 |
> | InfoXLM     | MRR@100    | 48.2 | 50.6 | 30.1 | 29.0 | 39.9 | 30.1 | 34.8 | 35.0 | 38.9 | 51.7 | 50.9 | 39.9 |
> |             | Recall@100 | 81.2 | 83.8 | 72.2 | 65.8 | 75.9 | 68.1 | 70.0 | 72.7 | 69.3 | 81.0 | 87.1 | 75.2 |
> | LaBSE       | MRR@100    | 50.1 | 52.3 | 29.7 | 41.3 | 48.3 | 27.6 | 33.4 | 37.3 | 54.6 | 56.7 | 43.6 | 43.2 |
> |             | Recall@100 | 83.0 | 85.6 | 71.4 | 80.2 | 86.1 | 63.2 | 67.0 | 74.3 | 86.7 | 89.4 | 81.9 | 79.0 |
> | XLMR-Long   | MRR@100    | 43.9 | 44.7 | 27.2 | 31.6 | 44.2 | 28.5 | 34.1 | 30.9 | 31.0 | 49.5 | 48.2 | 37.6 |
> |             | Recall@100 | 75.8 | 82.0 | 68.5 | 71.1 | 81.6 | 62.9 | 64.9 | 64.7 | 58.9 | 80.0 | 85.9 | 72.4 |
> | CROP        | MRR@100    | 46.2 | 39.9 | 27.7 | 34.0 | 47.0 | 26.2 | 32.1 | 31.8 | 41.5 | 55.9 | 46.3 | 38.9 |
> |             | Recall@100 | 80.3 | 84.2 | 70.9 | 73.8 | 85.5 | 63.1 | 68.1 | 70.4 | 72.0 | 86.3 | 85.9 | 76.4 |
> | MSM         | MRR@100    | 51.6 | 53.0 | 31.6 | 39.4 | 50.5 | 32.0 | 36.8 | 37.2 | 43.4 | 62.6 | 53.5 | 44.7 |
> |             | Recall@100 | 83.0 | 83.8 | 73.9 | 77.9 | 85.7 | 67.5 | 70.3 | 71.4 | 73.0 | 89.8 | 88.2 | 78.6 |
>
> For more detailed comparisons please refer to Sec.5.3 in our revised manuscript.
>
> Here we mainly clarify some misunderstandings and differences, as follows:
>
> a). MSM are trained without any supervised data, which keeps in line with mBERT and XLM-R. In contrast, mUSE, LASER and LaBSE are trained on text-matching datasets [1] such as parallel data, NLI data, QA data, etc, which provide more supervised signals. Indeed we also compared to LaBSE in the appendix of the original version, and we have added more baselines in the revised version.
>
> b). Our target is a universal pre-trained model, while some of the others [2, 3] focus on specialized (or fine-tuning) methods such as adapters [2], distillation [3], etc. And our pre-trained model MSM can be easily applied with these methods by replacing mBERT or XLMR in these papers.
>
> c). We follow the advanced setting that first fine-tunes and then evaluates for dense retrieval [7], which may differ from the previous methods of static word embeddings or fixed pre-trained models. This setting is suggested by the dataset paper of XTREME-R, Mr. TyDi and XOR, and the recent research [4, 5] also adopt this evaluation paradigm.
>
> d). mContriever [4] is trained on fewer languages and the corpus that is closer to the evaluation task, both of which can improve upon our setting, according to its ablation study [4]. That is because more languages face the curse of multilinguality and the closer corpus leads to better domain transfer. However, this setting may damage generalization and narrow the application scope. And there are several methods [4, 5, 6] similarly adopting cropped spans. So for a fair comparison, we have reproduced it as CROP under the same setting.
>
>
>
> References:
>
> [1] Litschko et al. ECIR 2021. https://arxiv.org/pdf/2101.08370.pdf
>
> [2] Reimers et al. EMNLP 2020. https://arxiv.org/abs/2004.09813
>
> [3] Litschko et al. COLING 2022. https://arxiv.org/abs/2204.02292
>
> [4] Izacard et al. TMLR 2022.https://arxiv.org/pdf/2112.09118.pdf
>
> [5] Wu et al. IJCAI 2022. https://arxiv.org/abs/2206.03281
>
> [6] Gao et al. EMNLP 2021. https://arxiv.org/abs/2104.08253
>
> [7] Ruder et al. EMNLP 2021. https://arxiv.org/abs/2104.07412

---

> ### Author Response · Authors · 2022-11-19
> **Response to Reviewer XpJe, Part 2**
>
> #### Response to Reviewer XpJe, Part 2
>
> > Q3. The paper also does conflate query-passage and sentence retrieval, and does not evaluate on document retrieval at all.
>
> Thanks for your valuable comments.
>
> a). According to [1], the document retrieval dataset CLEF gets even better results when truncated into 128 tokens than 256 tokens. And our evaluation dataset XOR and Mr.TYDI passages are 256 tokens long, which is comparable to the former. Though we mainly focus on passage retrieval, the outperforming results on XOR and Mr.TYDI can also demonstrate that our MSM can address different granularity of retrieval well.
>
> b). Besides, recent works on cross-lingual document retrieval [7, 8] are usually based on mBERT and XLMR and expose or split the documents to less than 512 tokens. Our MSM's input can be 512 tokens, and as a pre-trained model it's easy to be applied to replace mBERT or XLMR in the cross-lingual document retrieval papers [1, 8].
>
>
>
> > Q4. The paper also critically requires parallel data to work -> if one has parallel data, one of the must-have baselines are also MT-based query-translate or passage-translate approaches which sometimes/often work better than standard encoder-based approaches.
>
> a). In fact, MSM is trained without any parallel data and only utilizes multi-lingual raw text, and the setting of the pre-training corpus mainly keeps the same as XLMR.
>
> b). We have added MT-based baselines for cross-lingual retrieval in Appendix A.5. The results show that MSM outperforms Transformer-based MT and GMT (Google MT) is still a stronger method. However, MT-based methods depend on the quality of the MT system, and it relies on a two-stage pipeline which may lead to cumulative errors. In contrast, our MSM is easily applied with bi-encoder retrievers, which have more advantages in terms of deployments and diagnosis. For more details refer to Appendix A.5 in the revised version.
>
>
>
> > Q5. There are no discussions on how different target languages might impact the results: are all the languages equally difficult, which ones might cause major drops of performance and, most importantly, why?
>
> Thanks for the constructive advice. Please refer to Appendix A.2 in our revised manuscript, where we have added more analysis on multilinguality, summarized as follows: (1) MSM can achieve more gains in low-resource language. (2) The target languages closer to the pivot language usually perform better and achieve more improvements. (3) The multi-lingual data lead to better cross-lingual retrieval performance.
>
>
>
> Others that may be helpful:
>
> - We have added some citations and continually polished the paper.
> - As for the reproducibility, we will release the code and models soon.
>
> Thanks again for helping us make the paper stronger. Please let me know if you have any unclear things or additional concerns.
>
>
>
> References:
>
> [7] Yang et al. SIGIR 2022. https://arxiv.org/pdf/2204.11989.pdf
>
> [8] Nair et al. ECIR 2022. https://arxiv.org/abs/2201.08471

---

### Author Response · Authors · 2022-11-19
**Summary of Paper Revisions**

We thank all the reviewers for their constructive feedback, insightful comments, and generally positive appraisal of our work.

Our paper has been carefully revised according to the valuable comments, and the revised version has been submitted to OpenReview. The revised parts are highlighted in blue in our manuscript.  And the summary of the main revision is as follows:

1. We have modified some expressions and details to make the paper more clear, as suggested by the reviewers.

2. We have added more strong baselines in Sec.5.3, and provided more comparisons in Sec.2 and Appendix A.3.

3. We have added some other content in the appendix:

   a). more details of our pre-training data in A.1.

   b). more analysis on multilinguality in A.2.

   c). the comparison to translation-based method for cross-lingual retrieval in A.5.

---

### Decision · Program_Chairs · 2023-01-20

**Decision:**

Accept: poster

**Justification For Why Not Higher Score:**

More thorough comparisons with relevant related work.

**Justification For Why Not Lower Score:**

N/A

**Metareview: Summary, Strengths And Weaknesses:**

This paper presents a cross lingual representation learning method that uses the sequential information of sentences in parallel documents as a training signal. The topic is important and the proposed method makes sense.

The reviews are mixed. Reviewer XpJe thinks that there is a lack of comparison to major baselines and relevant related work in the area. Other reviewers have more positive recommendations about the paper, citing the simplicity of the method and the strong results as the basis for their scores.

There are concerns about whether the strong results of the proposed method is because the baseline is inadequate. However, I think the authors have sufficiently addressed the concerns of reviewer XpJe in their updated manuscript and author response. I recommend accepting the paper.

**Note From Pc:**

if the above contains the word "oral" or "spotlight" please see: "oral" presentation means -> notable-top-5% and "spotlight" means -> notable-top-25%. As stated in our emails, we are disassociating presentation type from AC recommendations